# Extra-Articular Distal Humerus Plate 3D Model Creation by Using the Method of Anatomical Features

**DOI:** 10.3390/ma16155409

**Published:** 2023-08-02

**Authors:** Nikola Vitković, Jelena R. Stojković, Nikola Korunović, Emil Teuţan, Alin Pleşa, Alexandru Ianoşi-Andreeva-Dimitrova, Filip Górski, Răzvan Păcurar

**Affiliations:** 1Faculty of Mechanical Engineering, University of Nis, Aleksandra Medvedeva, 18000 Nis, Serbia; jelena.stojkovic@masfak.ni.ac.rs (J.R.S.); nikola.korunovic@masfak.ni.ac.rs (N.K.); 2Department of Mechatronics and Machine Dynamics, Faculty of Automotive, Mechatronics and Mechanical Engineering, Technical University of Cluj-Napoca, Blv. Muncii, No. 103-105, 400641 Cluj-Napoca, Romania; emil.teutan@mdm.utcluj.ro (E.T.); alin.plesa@mdm.utcluj.ro (A.P.); alexandru.ianosi@mdm.utcluj.ro (A.I.-A.-D.); 3Faculty of Mechanical Engineering, Poznan University of Technology, Piotrowo 3 STR, 61-138 Poznan, Poland; filip.gorski@put.poznan.pl; 4Department of Manufacturing Engineering, Faculty of Industrial Engineering, Robotics and Production Management, Technical University of Cluj-Napoca, Blv. Muncii, No. 103-105, 400641 Cluj-Napoca, Romania

**Keywords:** distal humerus, LCP plate, parametric model, method of anatomical features

## Abstract

Proper fixation techniques are crucial in orthopedic surgery for the treatment of various medical conditions. Fractures of the distal humerus can occur due to either high-energy trauma with skin rupture or low-energy trauma in osteoporotic bone. The recommended surgical approach for treating these extra-articular distal humerus fractures involves performing an open reduction and internal fixation procedure using plate implants. This surgical intervention plays a crucial role in enhancing patient recovery and minimizing soft tissue complications. Dynamic Compression Plates (DCPs) and Locking Compression Plates (LCPs) are commonly used for bone fixation, with LCP extra-articular distal humerus plates being the preferred choice for extra-articular fractures. These fixation systems have anatomically shaped designs that provide angular stability to the bone. However, depending on the shape and position of the bone fracture, additional plate bending may be required during surgery. This can pose challenges such as increased surgery time and the risk of incorrect plate shaping. To enhance the accuracy of plate placement, the study introduces the Method of Anatomical Features (MAF) in conjunction with the Characteristic Product Features methodology (CPF). The utilization of the MAF enables the development of a parametric model for the contact surface between the plate and the humerus. This model is created using specialized Referential Geometrical Entities (RGEs), Constitutive Geometrical Entities (CGEs), and Regions of Interest (ROI) that are specific to the human humerus bone. By utilizing this anatomically tailored contact surface model, the standard plate model can be customized (bent) to precisely conform to the distinct shape of the patient’s humerus bone during the pre-operative planning phase. Alternatively, the newly designed model can be fabricated using a specific manufacturing technology. This approach aims to improve geometrical accuracy of plate fixation, thus optimizing surgical outcomes and patient recovery.

## 1. Introduction

In the field of orthopedic surgery, the utmost importance lies in delivering the best possible medical treatment for patients with bone fractures. Surgeons employ both internal and external fixation techniques to address these fractures [1,2,3,4,5]. External fixation is a technique that involves the utilization of a fixator positioned externally to the body to stabilize bone fragments [1]. On the other hand, internal fixation relies on the utilization of osteofixation materials such are screws, pins, and plate implants to stabilize the fractured bone [2,3,4,5]. Internal fixation is preferred due to its ability to promote superior functional recovery of the bone [2].

Plate implants are the most utilized internal fixation devices, and they are manufactured in various sizes and shapes to accommodate different patients [4]. However, a challenge arises when there is a geometrical mismatch between the size and shape of the patient’s bone and the standard plate implant [2,6,7]. In such cases, accurately positioning the plate becomes difficult, and the treatment may be complicated by inadequate load transfer during the bone healing process. To address this issue, personalized plate implants (PPI) have been introduced. These implants are specifically customized to align with the patient’s bone anatomy, meet the requirements of the surgeon, and ensure the preservation of blood flow in the periosteum during the recovery process [5,6,7]. While the implementation of patient-specific implants yields favorable outcomes for patients, it may necessitate additional time for pre-operative planning and manufacturing [7]. Consequently, patient-specific implants, such as the PPI, are employed when the use of predefined (standard) implants can potentially lead to complications during the intraoperative and postoperative phases. Creating an anatomically adjusted plate implant typically relies on a patient-specific bone model. Nevertheless, when it comes to fractures, the bone data acquired from medical imaging techniques can often be incomplete, lacking the necessary geometric and anatomical information required to construct a complete bone model. Various approaches and methodologies can be employed to develop a 3D model of the bone, utilizing either complete data (standard reverse modeling) or incomplete data (template and/or parametric models) [6,8,9,10,11,12,13].

In modern medicine, a variety of implants are used for the fixation of human bone fractures [14]. Dynamic Compression Plates (DCPs) with oval holes have been used to achieve inter-fragment compression using screws. These plates feature specially designed oval holes, such as those described in [15], which facilitate bone fragment compression upon screw tightening. The utilization of the Dynamic Compression Plate (DCP) offers numerous benefits, such as minimizing the occurrence of misaligned joints, providing stable internal fixation, and eliminating the need for external immobilization. Consequently, this allows for the mobility of neighboring joints while ensuring proper alignment and healing of the fractured bone. To ensure sufficient stability and support bone functionality, DCPs must be placed against the periosteum (the tissue covering the outer surface of bones) [14,15,16,17]. However, this requirement raises a crucial issue regarding the cortical bone’s porosity at the implant site due to limited blood supply. To minimize refracture risk, it is recommended not to remove the plate for at least 15–18 months to eliminate the gap between the bone fragments. Studies have analyzed the causes of refracture and concluded that cortical necrosis is a contributing factor [18,19]. A novel plate design, known as a limited contact-dynamic compression plate (LC-DCP), has been developed to address plate interference with cortical perfusion and reduce cortical porosity. The LC-DCP design involves less surface-to-surface contact with the bone’s periosteum compared to DCP, approximately 50% less. This reduction in contact aids in decreasing cortical bone necrosis and osteoporosis beneath the plate [16,17]. However, it should be noted that some studies [20,21,22] have shown that LC-DCP does not improve blood flow to the bone or the biomechanical properties of the bone-implant assembly. Locking Compression Plates (LCPs) have largely replaced the aforementioned implants in contemporary medicine. LCPs possess the ability to provide both locking and standard functions. Nonetheless, locked plating cannot entirely substitute conventional plating [14]. LCPs offer superior fixation and can withstand higher loads compared to standard plates such as DCPs [20,21]. Since LCPs do not require precise contouring and do not need to make direct surface contact with the bone when locking screws are used, they act more as fixators. However, the increased distance between the plate and the bone may present challenges [22,23,24,25]. Distal humerus fractures of the elbow represent around 2% of adult elbow fractures [22]. Extra-articular fractures, a specific type of distal humerus injury, require open reduction and adequate stabilization. In situations where bone quality is compromised, particularly in individuals with osteoporosis, the use of stable angular plates is essential. Thus, ensuring proper elbow stabilization during the patient’s recovery phase is of paramount importance [23,24,25]. To fulfill these needs, Locking Compression Plate (LCP) extra-articular distal humerus plates are commonly employed as implants for the treatment of this particular fracture. These plates offer fixation systems that not only conform to the anatomical shape of the bone but also provide angular stability [26]. It is crucial to closely monitor and regulate the contour of the LCP plate as it conforms to the distal portion of the humerus. The distal end of the plate should exhibit a curved shape along the posterior aspect of the lateral column, while ensuring that the plate remains a safe distance away from the olecranon fossa, thereby allowing for unimpeded complete extension of the elbow joint. In certain clinical cases, the LCP plate needs to be bent during surgery [7,23,26]. The need for intraoperative bending can be avoided by utilizing predefined Personalized Plate Implants (PPIs). To gain a better understanding of plate implants and their potential applications within the human body, it is essential to comprehend various types of standard biomaterials employed in their production. Bio-metals, which are inorganic metallic biomaterials, are the most used biomaterials in medical practice for fabricating medical devices such as bone plates [27]. Bio-metals are typically non-biodegradable; however, ongoing research aims to develop alloys with biodegradable properties. Stainless steel, cobalt alloys, and titanium alloys are the primary bio-metals used in orthopedic surgery. However, bio-metals can cause corrosion-related toxicity and early failure due to heterogeneous stress distribution since implant modules are more elastic than bone. As a result, several studies [27,28,29] have investigated the potential of organic biomaterials and bio-composites, including Polymethyl-Methacrylate (PMMA), Poly Lactic Acid (PLA), Poly Glycolic Acid (PGA), Polydioxanone (PDS), and others. These studies have provided evidence that orthopedic implants constructed from biopolymers and bio-composites can serve as viable alternatives to metallic implants. Currently, these materials find extensive use in dental implants and smaller, flexible internal fixators.

### The Proposed Solution for Creating a 3D Model of the Plate

The initial modelling of the plate contact surface model was developed and presented in [1] (Manufacturing 2022 conference). To improve the PPI model, additional research work has been conducted, which is presented in this paper. In [1], two different methodologies were demonstrated for the creation of the plate surface model. The first methodology is based on the spline curves application and the second was based on the application of SubD patches. The presented results were more than promising; therefore, in this study, additional improvements of these two methodologies were made, and a new approach for developing a completely novel parameterized parametric 3D model of the plate is introduced. The main improvements are reflected in the application of the Characteristics Product Features (CPF) methodology [30], which is an essential part of the improved Method of Anatomical Features (MAF) [6]. CPF and MAF were created by the authors of this paper, and they can be used as novel tools for human organ remodeling and different applications (education, additive manufacturing, clinical implementation). Using these improved MAF, it is possible to define more requirements that the resulting model should fulfil, thus making it better in different ways (geometrically, anatomically, and technologically). As a novel addition, the new parametric model should bring the possibility to create 3D plate model (solid) even when there is not enough bone data to provide direct bone remodeling, defined in the first two methodologies, thus making it a useful tool for real clinical applications in pre-, intra-, and post-operative procedures. The improved PPI has additional geometrical elements that can be used as the basis for remodeling, thus improving pre-operative (software bending and transformations), intra-operative (bending during surgery is lesser because of better models), and post-operative (better recovery because of shortened operation time, better fitting of the plate) processes.

## 2. Materials and Methods

The MAF introduces a novel approach for describing the geometric characteristics of human bones, allowing for the creation of diverse geometric models. Two types of models can be generated using MAF:3D Geometrical Models: These models, which have been used in Computer-Aided Design (CAD) for many years, encompass standard polygonal, surface, and volume models. CAD software (CATIA V5) package was employed to create these models by applying conventional CAD technical features.Predictive (Parametric) Models: By utilizing morphometric and other readable parameters obtained from medical imaging techniques, MAF can adjust these models to match the geometry and morphology of a specific patient’s bone. These models are based on input data acquired from medical imaging methods such as CT or MRI.

Both models rely on data acquired from medical imaging methods and are created through the MAF application, which consists of basic and additional processes. The basic processes [6] enable a comprehensive geometric and anatomical definition of the human bone. These processes include: creation of the initial polygonal model: this process produces a polygonal model representing the specific human bone; anatomical analysis: this process generates an anatomical model of the specific human bone; definition and/or selection of Referential Geometrical Entities (RGEs): these entities serve as fundamental geometry elements used in the creation of other geometric elements such as curves or surfaces; creation of Constitutive Geometrical Entities (CGEs): these entities, known as constitutive entities, are used in the creation of surface and solid models of human bones and bone parts. The polygonal model, RGEs, and CGEs are outputs of the basic MAF processes and can be used to create different geometrical models of the human bone. The additional processes in MAF produce a parametric model specific to the human bone. Based on predictive functions, this parametric model defines the shape and anatomy of the bone and is defined by the values of morphometric parameters acquired from medical imaging methods. The parametric model can be transformed into a personalized model by applying morphometric parameter values obtained from a specific patient’s medical images. The effectiveness of MAF has been demonstrated through various studies [6,7,19], which have shown its geometrical accuracy as well as its anatomical and morphological correctness when applied to human bones. Further examples of MAF application can be found in chapters [10,11,12] of the book “*Personalized Orthopedics*” [31] published by Springer. In addition to bones, MAF has been utilized in the creation of personalized implants for human bones, including long and flat bones. Ethical regulations have been and will continue to be followed for all real (clinical) applications.

The Characteristics Product Features (CPF) expands the capabilities of MAF by enabling the definition of specific features of human organs beyond anatomical characteristics. This allows the resulting model, called the Features Model (FM) or specific CPF model, to possess additional properties such as functional, material, manufacturing, or mechanical characteristics. It is important to note that in this context, the term “model” refers to a complex entity comprising different elements (CGE, polygonal model(s), CPF properties). The FM resulting from the application of MAF can meet various needs and fulfill different requirements from different stakeholders. The FM can serve as an input model for additional MAF processes, facilitating better parameterization according to the novel specifications defined in the FM, such as material or manufacturing properties. Figure 1 illustrates the relationship between the MAF model and the integration of CPF into the methodology itself. The CPF is a methodology already used in CAD/CAM to define parametric models of different mechanical parts [30]. For this application, if the requirement is to make the bone or plate model more geometrically accurate, then CPF can help the designer and physician to determine the anatomical area(s) where this requirement needs to be accomplished and to mirror the known morphology and anatomy of the model in hand or use tacit knowledge to apply different (more accurate) geometrical elements. Overall, the MAF and CPF methodologies offer a comprehensive approach for creating geometric models of human organs, allowing for personalized and customizable features.

### 2.1. Creation of PPI LCP Models

The two initial approaches for the plate contact surface creation are presented in [1], and they are as follows:The Curve Based Method (CBM) that utilized commonly used geometric elements (spline segments) as parametrized geometry. The bone model was then employed as a deformable template model through the application of Free-Form Deformation (FFD) techniques.The Subdivision Surface-Based Method (SDBM) and Regions of Interest (ROI) were employed to define a control mesh, acting as the parameterized grid. By adjusting the vertices and normals of the control mesh polygons, the SubD surface was capable of conforming to the underlying bone template model.

The newly developed approach/methodology is based on innovative MAF, and it is defined as follows:The CPF creation method—The produced FM model is based on the additional product characteristics/properties, not just geometrical and anatomical. The additional properties are defined as parameters, including shape modification, geometry and topology optimization, material selection, and manufacturing process definition. To describe and define additional properties, different descriptive elements can be used. For example, if we need to impose some limitations to plate manufacturing, a simple programming script can be added to the programming routines. This programming script can be used as a control mechanism.

The new Feature Model (FM) was introduced to improve the first two methodologies and enable a better parametrization of an optimized PPI model. The FM model is a direct output of CPF application, and it brings additional capabilities to PPI creation:(a)The material model, closely related to the manufacturing model, enables the definition of adequate material for plate production using additive manufacturing.(b)The production capability reflects the possibility of plate model manufacturing. In this study, additive manufacturing was chosen as the primary methodology for producing PPI. Additional manufacturing technologies, such as cutting or forging, will be considered in future work.(c)The improved geometrical model includes additional ROI points to enable better capability for shape modification. The original models [1] were accurate enough, but extra ROI points were selected to improve model capability for geometrical/shape adaptation to the specific bone. This FM feature is very useful in pre-operative surgery simulation [6,7].(d)The formation of the workflow system for the application of PPI in clinical or other cases (e.g., in medical education). This workflow model is presented for the optimized parametric model application.

The methods employed in this study utilized a pre-existing surface model of the left humerus [1,19]. This model was generated from a CT scan of a 50-year-old woman using a Toshiba Aquilion 64-slice scanner at the Clinical Center Nis. The CT scan had a resolution of 512 × 512 pixels and a scan thickness of 0.5 mm. The CAD program CATIA (licensed, V5) was used for the designing process (Figure 2a). In Figure 2b, the surface model of the humerus shaft and distal part [1] and original trimmed spline curves are shown. An extra curve was also introduced in the distal section (condyle) to improve geometric accuracy and anatomical correctness based on the surgeon’s recommendation. A descriptive model defined in the literature was introduced to simulate the extra-articular fracture in the distal humerus, classified as A2 type according to OTA/AO classification [32]. The visual representation of this simulated fracture can be observed in Figure 2c.

### 2.2. The Improved Curve Based and Subdivision Surface-Based Methodologies

Accurately determining the position and orientation of the standard LCP plate was of utmost importance to establish a valid plate–humerus contact surface parametric model. This was accomplished by referring to literature [33] and considering recommendations from physicians. Specific spline curves were created by trimming existing curves [1], as shown in Figure 3a. These curves served as the basis for constructing the PPI LCP contact surface model. Using this model, a volume (solid) model of the plate was generated by adding a thickness of 2 mm. The initial guidelines aimed to encompass a broader area in the distal part of the humerus and ensure a more anatomically tailored fit. Furthermore, it was essential to minimize contact with the Olecranon fossa while ensuring that the plate’s distal part remained as close as possible to the extension of the shaft part of the contact surface. For the SubD surface implementation, the original mesh was defined by utilizing the created curves and the original bone surface, as shown in Figure 3b. Subdivision surfaces are mesh-based objects that are well-suited for approximating models such as character modeling and creating smooth organic forms, making them particularly useful for real-time model adaptation. This study defined parametric patches across the humerus surface model using all the specified curves and points on them. This approach ensures the preservation of the bone’s inherent shape while adhering to Subdivision (SubD) surface modeling principles. For the SubD surface, if more control points are defined, as it is in theses case, mesh can be more adaptable, and thus the designer has more flexibility for the plate design. Both model types were developed in [1], with good anatomical and geometrical accuracy (median deviation around 0.1–0.2 mm). The overall conclusion was that achieved accuracy is more than enough, but as it was stated, the physicians had additional requirements that were mainly focused on the possibility of making the model more usable. To do that, two main improvements were defined. The first improvement was to add additional curves (in horizontal, transversal direction) for direct surface creation, and an additional spline curve as the surface spine (coronal direction).

The additional curves and spine curve are presented in Figure 3c. The total number of added transversal curves was five, three in the distal/shaft section, and two in the condyles section (already used for bone surface model improvement, only trimmed). The newly developed surface model based on original and added curves is presented in Figure 3c. The deviations between models and bone surface were not changed drastically, and they remained in the already achieved boundaries. The deviations for the new curve model are presented in Figure 3d–f and for the novel SubD model in Figure 3g–i. It can be concluded that both models are still very geometrically accurate and anatomically correct, i.e., deviation maximums were reduced by about 5% for both models, in the end being evenly distributed. One thing that has changed is the number of points used for ROI definition [1], which is now greater because of the additional curves. The designer now has an option to modify more points, as well as to adjust the plate shape model accordingly. The modified model was adjusted by manually manipulating the point coordinates, thereby altering the overall geometry of the plate shape. To ensure compatibility with the patient’s bone anatomy and shape, the plate model can be personalized and scaled using X-ray images, including anterior–posterior or lateral–medial views, or a combination of both [1]. The points that define the plate model can be manually adjusted through various transformations. Alternatively, the PPI LCP plate model can be created and utilized as a template in order to bend the standard LCP plate model during surgery. Additionally, the PPI model can undergo further processing to incorporate locking and regular holes, resulting in a CAD model suitable for Computer-Aided Manufacturing (CAM) processes such as milling and casting.

### 2.3. CPF Methodology for the Creation of Additional Models and Process Workflow

The final approach includes optimizing the plate model geometry and creating the parametric plate model, material model, and manufacturing model. The Parametric Plate Model (PPM) is formed on the bases of the improved geometry, and it is defined as a model for which geometry can be changed by changing the parametric values, thus making it personalized to specific patent. Material and manufacturing models are defined as material properties of selected material and technological process of chosen manufacturing technology. These models are essential to the workflow for creating and applying the plate model in surgical cases. The simplified workflow model is presented in Figure 4.

The first process is medical imaging of the patient (P1) in which parameter values are acquired. Next, the application of these values into the parametric model is performed (P2), which enables the modification of the parametric surface/solid model into the patient-specific 3D plate model. The next step is to analyze the resulting model and make additional improvements if necessary (P3). When the physician and designer are satisfied with the model, the material and manufacturing model formation process follows (P4). Finally, the complete (Feature) model (P5) is defined and ready for application, possibly for manufacturing, presentation, simulation, and education. It is important to note that the final geometrical model is created at the end of the process because the manufacturing model may require some geometry changes to be correctly produced. For this workflow to succeed, it is possible to use all defined models, but in this case, the definition of the third (optimized PPM) is presented. The optimized PPM is also based on the same bone geometry but with additional adjustments, and to create it, several steps are required:Creation of the optimization curves that can be easily adapted. Circular arcs are used as the simple geometrical element defined by three points: Left max, Right max, and Middle point of the corresponding spline curves (used in first two methods). According to the previous testing, nine cross-sectional curves provide the plate model with enough accuracy, so nine circular arcs are defined. These three points are specific ROIs defined on each spline, as shown in Figure 5.The definition of the left and right boundary guiding curves going through left and right maximum points on each spline curve, as shown in Figure 5.


Definition of the angles of the circular arcs. An important parameter for the easier control of the plate width in cases when boundary-guiding curve application cannot produce a manifold model, or medical images do not provide enough data.Definition of the circular arc center distance to the bone anatomical axis as parameters [7], as shown in Figure 6. This is necessary for cases where there is only one X-ray image available. By using these distances and arc angles, a less accurate plate model can be created, but it is still usable for additional processing.Formation of the matrix that includes ROI points for each parametric curve (circular arc) and for the left and right guiding curve (1)—*Opt_ROI_*.Formation of the vector of circular arc angles for each section (1)—*Opt_angles_*Formation of the vector of arc middle point distances to the anatomical axes (1)—*Opt_dist_*Definition of additional or supporting distances for better calculation of plate geometry. Supporting distances are measured for the left and right circular arc point, and they are defined as normal distances from these points to the anatomical axes in the AP (Anterior–Posterior) plane (1)—*Opt_Sdistl/r_* (d_1l/r_ means d_1l_ (left) and d_1r_ (right) distances to points—these distances form two vectors).

(1)
$${Opt}_{ROI}=\left[\begin{array}{c} \begin{array}{ccc} L_{11} & M_{12} & R_{13}\\ L_{21} & M_{22} & M_{23} \end{array}\\ \cdots \\ \begin{array}{ccc} L_{91} & M_{92} & R_{93} \end{array} \end{array}\right];{Opt}_{angles}=\left[\begin{array}{c} \begin{array}{c} \propto_{1}\\ \propto_{2} \end{array}\\ \ldots \\ \propto_{9} \end{array}\right];{Opt}_{dist}=\left[\begin{array}{c} \begin{array}{c} \begin{array}{c} d_{1}\\ d_{2} \end{array}\\ \ldots \end{array}\\ d_{9} \end{array}\right];{Opt}_{Sdistl}=\left[\begin{array}{c} \begin{array}{c} \begin{array}{c} d_{1l}\\ d_{2l} \end{array}\\ \ldots \end{array}\\ d_{9l} \end{array}\right];{Opt}_{Sdistr}=\left[\begin{array}{c} \begin{array}{c} \begin{array}{c} d_{1r}\\ d_{2r} \end{array}\\ \ldots \end{array}\\ d_{9r} \end{array}\right]$$

Parameter that defines the thickness of the plate solid model. This parameter is defined according to the physician specifications, and usually the value is 2 mm, but it can be set up as required.


**Figure 6 materials-16-05409-f006:**
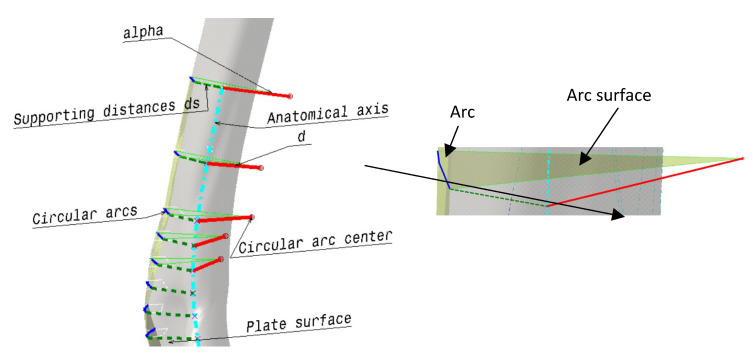
The additional geometry for the definition of the parametric plate model.

These four elements presented in (1) define the optimized parametric model of the plate surface, which can be used when the parametric bone model is not defined, there are not enough medical images available, or the images are of poor quality. Even where no medical image is available, the initial plate model can be used as a template and then bent to the requirements of the specific case (e.g., during surgery), thus making it personalized. In a case of poor-quality or non-uniform X-ray image, bone and plate geometrical models can be used as FFD templates. Surgeons can use these models to tailor the plate to the patient bone by scaling to the overall X-ray image in AP or normal plane during pre-operative planning. It is important to note that for adequate reading of the defined parameters during X-ray imaging, it is important to use etalon for dimension scaling, or to use the bone FFD model to scale the dimensions. The good thing is that when the parametric humerus model is finished, the workflow can include both models (plate and bone) for complete process automation, from medical imaging to manufacturing process definition.

The material model is the simplest, and it refers to material selection and the definition of its characteristics. This model is defined by the purpose of the resulting manufactured model, i.e., if it is for presentation or education, then plain PLA, ABS or PA material can be used for plastic prototypes [34,35], but if it is for the actual application in a specific clinical case, then titanium alloys [36] or stainless steel [37] can be defined with different additive technologies such as Selective Laser Sintering (SLS), Selective Laser Melting (SLM), Electron Beam Melting (EBM), etc. The manufacturing model follows the material model, and it is the final model defined in CAM software. This refers to the specifications designated for 3D printer slicer or G-Code for cutting.

The defined parametric model was used with the patient’s X-ray image described in [1] (anonymous data, from Clinical Center Nis, only for testing and presentation purposes). The procedure for PPI application to create personalized plate model follows:The defined PPI model was adapted (scaled) to fit the dimensions of the scanned bone.Instead of etalon, the FFD bone model with known dimensions was scaled to the X-ray image in the AP (Anterior–Posterior) plane. For the initial testing, this is quite satisfactory.The ROI points were collected, and distances were calculated.The angles for circular arcs were applied to define curves lengths.The personalized plate surface model was created.A thickness of 2 mm was added and plate model was created, as shown in Figure 7a.

As a potential methodology for production FFF printing was chosen. Cura software (v5, free software) was used as a slicer for FFF printing. Considering the available technologies, the plate and bone models were printed using CreatBot DX Plus 3D printer (FFF—, ABS, infill 90%, layer thickness: 0.1 mm, with support structure), as shown in Figure 7b,c. The material was PLA, and production time was 13 h 5 min for bone, and 1 h 10 min for plate, very close to the printing time demonstrated in [1], which is another proof of the method validity. The plate model was fine-tuned to align with the shape and anatomical characteristics of the bone. In situations where a metal implant is required, alternative additive technologies such as Direct Metal Laser Sintering (DMLS) can be utilized. DMLS allows for the precise laser sintering of metal powder, providing a suitable method for fabricating the implant. Alternatively, conventional machining techniques such as forging can be employed. These production processes enable the customization of plate implants, facilitating their direct insertion into the patient’s body during surgery. This integration of pre-operative planning and surgical intervention with personalized plate implants significantly contributes to improved patient recovery outcomes.

## 3. Conclusions

In this paper, two approaches, curve-based and SubD meshing, for developing an PPI LCP extra-articular distal humerus plate model were improved and presented together with a procedure for developing and applying the optimized parametric plate model. Both approaches enable the creation of a surface and solid plate model, which can be adjusted to a specific patient’s bone. The idea of this study was not to compare these methodologies to the existing reverse engineering techniques, because if enough work is applied, these models can be formed with near zero deviation, but to show the possibility for geometrical deformability of the resulting models and their adaptation to the specific bone. Both improved methods are completely applicable at the current research stage, as demonstrated in the paper. The PPI LCP plate model can be modified and customized using these processes to align with the specific bone geometry and anatomical needs of the clinical case. This ensures that the plate implant is precisely tailored to meet the requirements of the individual patient. The formation of an optimized parametric plate model is presented to show how one geometrically simpler model can be formed and still used in education or clinical situations but, with additional consideration concerning usability, i.e., possible further bending during surgery may be required. In future work, a significant improvement will be implemented by introducing a parametric humerus bone model. This model will allow the materialization of a patient-specific (personalized) bone model. By incorporating this additional component into the methodology, the contact surface model will be fully and automatically adapted to the patient’s unique bone structure. Consequently, an optimized parametric plate model can be generated, resulting in enhanced geometrical and anatomical accuracy. The methodology can and should be applied to soft tissue remodeling, which is a very complex and dynamic structure. To enable soft tissue remodeling, we must add different properties, i.e., use CPF to define the complex model of the “complex” soft tissue entity. The presented advancements and potential for future upgrades can improve patient treatment and facilitate a smoother recovery process.

## Figures and Tables

**Figure 1 materials-16-05409-f001:**
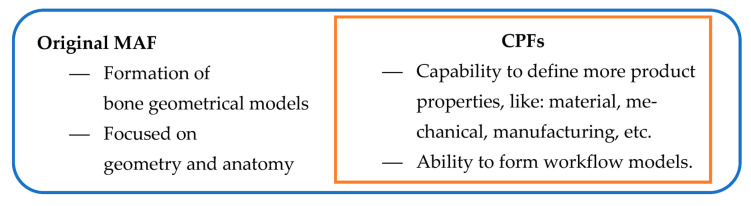
The MAF with CPF integration.

**Figure 2 materials-16-05409-f002:**
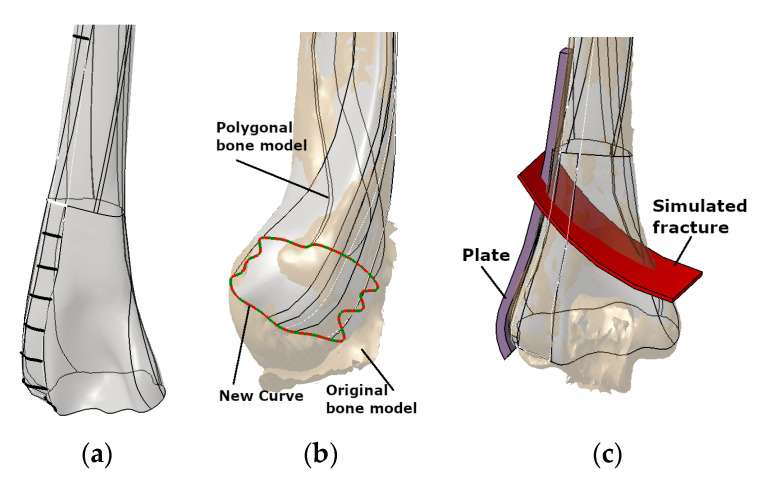
The definition of the LCP plate, along with the bone and fracture models, is as follows: (**a**) humerus surface model integrating trimmed anatomical curves (CGEs), (**b**) new curve on the bone model, and (**c**) simulated fracture demonstrating the placement of the LCP plate [1].

**Figure 3 materials-16-05409-f003:**
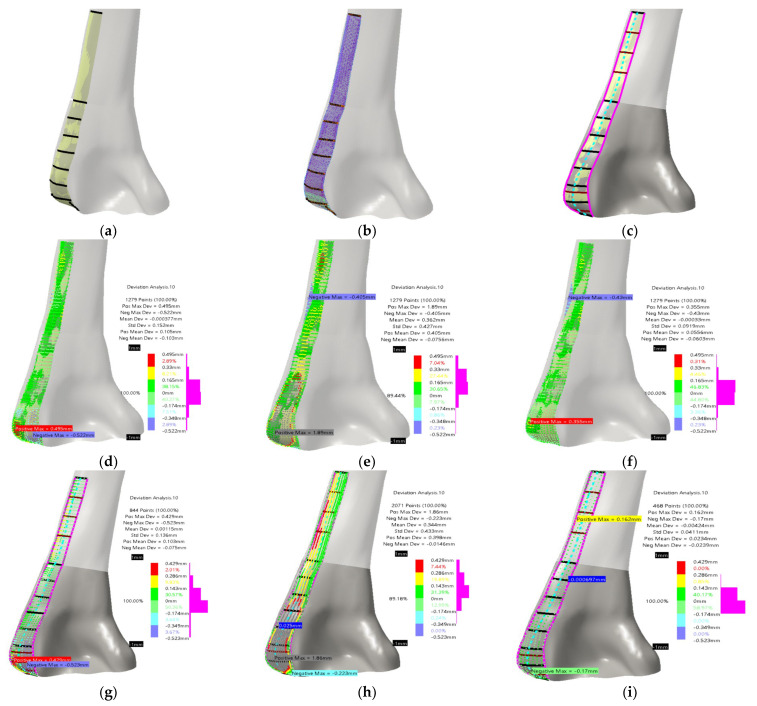
Definition of previous [1] and new improved LCP plate surface and with bone model: (**a**) original curve based model; (**b**) original SubD model; (**c**) additional curves and new plate surface model; (**d**) deviation analysis between optimized plate surface model and new curve based surface; (**e**) deviation analysis between SubD plate surface model and new curve based surface; (**f**) deviation analysis between curve based plate surface model and new curve based surface; (**g**–**i**) same deviations defined with the new SubD model.

**Figure 4 materials-16-05409-f004:**
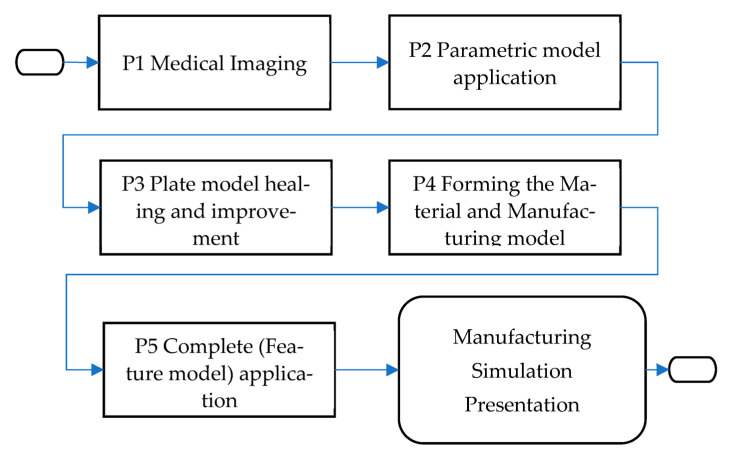
The simplified workflow model (only processes) of the optimized parametric plate model application.

**Figure 5 materials-16-05409-f005:**
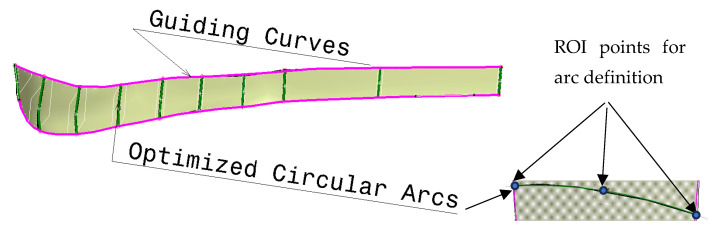
The optimized cross sections (circular arcs) and guiding curves for parametric plate model definition.

**Figure 7 materials-16-05409-f007:**
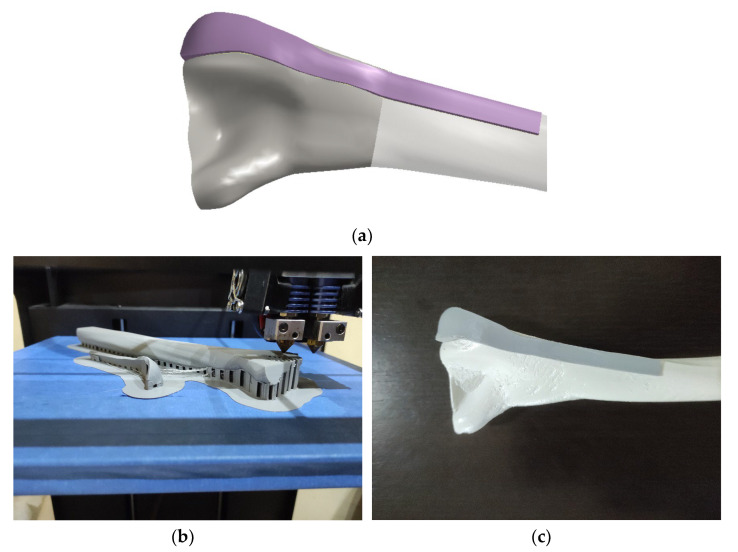
Prototype production: (**a**) personalized 3D plate solid model; (**b**) 3D printing process; (**c**) bone and plate model assembly.

## Data Availability

Not applicable.

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
