# Peer review of "Extra-Articular Distal Humerus Plate 3D Model Creation by Using the Method of Anatomical Features"

_materials, 2023, doi:10.3390/ma16155409_

Round 1

Reviewer 1 Report

The manuscript entitled "Extra-Articular Distal Humerus Plate 3D model creation by using the Method of Anatomical Feature" proposed an Anatomical Features (MAF) in conjunction with the Characteristic Product Features methodology (CPF) to enhance the accuracy of plate placement during surgery, which is significant. The proposed feature model and optimization method can be seen to help aid designers in providing more reliable bone support.

1. Fig. 3 (e,h) is partially covered by Fig.3 (f,i); some units disappeared.

2. Can the authors elucidate more (e.g., parameters used) on the X-ray-generated images since different X-ray scanning speeds and X-ray voltages can lead to distinct bone image quality?

3. In equation(1), can the author explain why r is the same for different d?

4. Most of the citations are pretty old with no DOI inserted; please correct the format and add more recent relevant citations to show the current progress on this topic.

Author Response

1. Fig. 3 (e,h) is partially covered by Fig.3 (f,i); some units disappeared.

Corrected. Thank you!

2. Can the authors elucidate more (e.g., parameters used) on the X-ray-generated images since different X-ray scanning speeds and X-ray voltages can lead to distinct bone image quality?

The methodology can use two approaches.

  1. The first approach refers to good quality X-ray and etalon which determines the dimension scaling.
  2. The second approach refers to using the existing plate and bone models as templates and scaling the model in the AP plane to the overall bone dimensions. This approach can be used when the X-ray is non-uniform, or some other artefacts exist. This approach refers to your question.

We added extra clarification in the text.

 3. In equation(1), can the author explain why r is the same for different d?

We corrected that. It is not the same r, yet it is left and right point. For example, d1l/r means d1l (left) and d1r (right) distances. These distances form two vectors.

4. Most of the citations are pretty old with no DOI inserted; please correct the format and add more recent relevant citations to show the current progress on this topic.

We added new citations to show the recent achievements, but some old citations need to stay because of the excellent introduction to the different topics and methodologies. They can help the reader to understand the topic of the paper.

Reviewer 2 Report

The authors discuss the importance of proper fixation techniques in orthopedic surgery, specifically for treating distal humerus fractures, typically using Dynamic Compression Plates (DCP) or Locking Compression Plates (LCP). To enhance precision and surgical outcomes, the study introduces the Method of Anatomical Features (MAF) and the Characteristic Product Features (CPF) methodology, enabling the development of a patient-specific parametric model for the contact surface between the plate and the humerus, which can either guide the bending of a standard plate pre-surgery or be used to fabricate a custom one.

A few minor points should be addressed before it can be accepted for publication:

- The introduction is really long. I suggest it may be shortened.

- Can you clarify how the Characteristics Product Features methodology (CPF) and the Method of Anatomical Features (MAF) work together to create a more precise model?

- How does the enhanced MAF allow for more detailed requirements to be set for the model, and can you provide examples of these requirements?

- You mention that the new parametric model can create a 3D plate model even with insufficient bone data. Could you expand on the technology or methodology that enables this? How significant is the improvement of the model in terms of surgical outcomes when using this newly improved plate model?

- You mentioned that CPF and MAF can be used for "human organ remodeling and different applications". Can you elaborate on these other potential applications? What are the potential limitations of your proposed approach, and how do you plan to address these in future studies?

- Given that each individual's bone structure is unique, how do these methodologies ensure the production of a perfectly fitting plate? How does the creation of these plate models using the improved methodologies influence the overall time and cost of the surgical procedure? In what ways does this enhanced modeling approach improve pre, intra, and post-operative procedures?

The Manuscript reads well, only a few minor mistakes were found across the text.

Author Response

The introduction is really long. I suggest it may be shortened.

We modified the text as much as possible due to the respect to other reviewers.

Can you clarify how the Characteristics Product Features methodology (CPF) and the Method of Anatomical Features (MAF) work together to create a more precise model?

The CPF is a methodology already used in CAD/CAM to define parametric models of different mechanical parts. It is not the main point to make it more geometrically accurate yet to make it more usable and practical in medical applications.  If the requirement is to make the model more accurate, then CPF can help the designer and physician to determine the anatomical area where this requirement needs to be accomplished and to mirror the known morphology and anatomy of the model in hand or use tacit knowledge to apply different (more accurate) geometrical elements. 

How does the enhanced MAF allow for more detailed requirements to be set for the model, and can you provide examples of these requirements?

To put it simply, we use different descriptive elements to define additional properties of the model. For one model in manufacturing, we added a simple programming script to define the limitations of the machine (because of some broken parts). This programming script is used as a control mechanism. In this case, we used simple manufacturing and material properties as stated in the text. 

You mention that the new parametric model can create a 3D plate model even with insufficient bone data. Could you expand on the technology or methodology that enables this? How significant is the improvement of the model in terms of surgical outcomes when using this newly improved plate model?

The model scalability and deformability is improved. The reference in the text to MAF demonstrates te application of the bone parametric models to create personalized models of the bones or plates. The bone parametric model is a scalable model and if enough parameters are acquired, then it can be used, but if a small number of parameters exists or there is not enough data to use the parametric model, then we can use the existing bone model to scale it as much as possible (e.g., to the X-ray), i.e. used it as FFD model.

You mentioned that CPF and MAF can be used for "human organ remodeling and different applications". Can you elaborate on these other potential applications? What are the potential limitations of your proposed approach, and how do you plan to address these in future studies?

Soft tissue is a very complex and dynamic structure, and to enable modelling, we must add different properties, so CPF will help to define the complex model of the "complex" entity in future work. 

Given that each individual's bone structure is unique, how do these methodologies ensure the production of a perfectly fitting plate? How does the creation of these plate models using the improved methodologies influence the overall time and cost of the surgical procedure? In what ways does this enhanced modeling approach improve pre, intra, and post-operative procedures?

In orthopedy, in the general case, we do not need the perfectly fitted plate because of possible necrosis and bone self-recovery, but in a geometrical sense, if we have an excellent input bone model, then it is possible to create a "perfect" plate. Considering the mathematics behind it (NURBS, SubD, meshes), it will always be some tolerance.  

The main benefits are for the patient - shortened surgery time and better plate adaptation. The cost of the procedure can vary. The standard plates are cheaper because they are made in series, yet this is personalized, which means one production unit. The main conclusion is that the surgeon defines the better solution for the patient. 

The model is improved in a sense that there are more geometrical elements which can be used as the basis for remodeling, thus improving preoperative  (software bending and transformations), intraoperative (bending during surgery is lesser because of better models) and postoperative (better recovery because of shorten operation time, better fitting of the plate) processes.